# Glycyrrhizinic Acid as an Antiviral and Anticancer Agent in the Treatment of Human Papillomavirus

**DOI:** 10.3390/jpm13121639

**Published:** 2023-11-24

**Authors:** Victoria Bravo, María Serrano, Alfonso Duque, Juan Ferragud, Pluvio J. Coronado

**Affiliations:** 1Gynecology and Obstetrics Service, Hospital 12 de Octubre, 28041 Madrid, Spain; 2Gynecology and Obstetrics Service, Hospital la Paz, 28046 Madrid, Spain; 3Gynecology and Obstetrics Service, Hospital Ruber Internacional, 28034 Madrid, Spain; 4Medical Department, Atika Pharma, 35002 Las Palmas de Gran Canaria, Spain; 5Women’s Health Institute, San Carlos Clinical Hospital, dISSC, Complutense University, 28040 Madrid, Spain

**Keywords:** glycyrrhizinic acid, human papillomavirus, antiviral, anticancer, antiproliferative, cervical lesions, LSIL, cervical cancer, condyloma, multifocal epithelial hyperplasia

## Abstract

Human papillomavirus (HPV), like any other virus, needs to penetrate the host cell and make use of its machinery to replicate. From there, HPV infection can be asymptomatic or lead to benign and premalignant lesions or even different types of cancer. HPV oncogenesis is due to the ability of the viral oncoproteins E6 and E7 to alter the control mechanisms for the growth and proliferation of host cell. Therefore, the use of agents with the ability to control these processes is essential in the search for effective treatments against HPV infections. Glycyrrhizinic acid (Gly), the active ingredient in liquorice, has been shown in numerous preclinical studies to have an antiviral and anticancer activity, reducing the expression of E6 and E7 and inducing apoptosis in cervical cancer cells. In addition, it also has antioxidant, anti-inflammatory, immunomodulatory or re-epithelializing properties that can be useful in HPV infections. This review includes the different antiviral and anticancer mechanisms described for Gly, as well as the clinical studies carried out that position it as a potential therapeutic strategy against HPV both through its topical application and by oral administration.

## 1. Introduction

Human papillomavirus (HPV) infection affects both women and men and can cause both benign lesions (warts or condylomas) and precancerous lesions and even some types of cancer such as cervical, anal, vaginal, vulvar, penile and oropharyngeal cancers [1].

Malignant transformation of HPV-infected cells is due to the expression of E6 and E7 oncoproteins that lead to the degradation of p53 and pRb and stimulate entry into the S phase without G1 arrest. As a result, cells acquire mechanisms to evade programmed cell death (apoptosis) and uncontrolled cell proliferation occurs, resulting in carcinogenesis [2].

Some antivirals have been evaluated for the treatment of HPV, but there are currently no antiviral drugs marketed for this indication. Current treatments for benign or premalignant HPV lesions are based on excisional or destructive treatments, as well as cytotoxic agents (e.g., podophyllotoxin) or immunomodulators (e.g., imiquimod). HPV-derived cancers are treated with surgery, radiotherapy, chemotherapy or immunotherapy. However, the treatment of HPV-associated problems and the prevention of HPV-induced malignancy requires new agents that provide new benefits such as dual antiviral and anticancer action.

Glycyrrhizinic acid (Gly), also known as glycyrrhizin or glycyrrhizic acid, is a natural triterpene saponin found in the root and rhizomes of plant species of the genus Glycyrrhiza *(Glycyrrhiza glabra*, *G. uralensis*, etc.), popularly known as liquorice. In humans, orally administered Gly is largely hydrolysed prior to absorption by intestinal bacteria, leading to glycyrrhetinic acid or enoxolone. Both glycyrrhizinic acid and glycyrrhetinic acid have shown similar antiviral and anticancer physiological properties [3,4]. Intravenous administration of Gly has also been evaluated, although toxicity is higher than through oral administration [5]. This makes both local and oral administration of Gly potentially useful for the treatment or prevention of viral infections or cancer and may be especially useful in problems resulting from HPV infection (Figure 1).

## 2. Antiviral Activity of Glycyrrhizinic Acid

Numerous in vitro and in vivo studies suggest that Gly is a broad-spectrum antiviral against different viruses affecting both humans and animals. Among the viruses affecting humans for which Gly’s antiviral action has been observed are, in addition to human papillomavirus (HPV), herpesviruses (HSV-1, HSV-2, VZV, EBV, KSHV), hepatitis viruses (HBV, HCV), influenza viruses, human immunodeficiency virus (HIV), coronaviruses (SARS-CoV, SARS-CoV-2), rotavirus, coxsackievirus or Zika [4,6,7].

The mechanisms of antiviral action of Gly are multiple and may differ among viruses, and the exact mechanisms are still unknown in some cases. For example, in the case of oncogenic viruses such as HPV, Kaposi’s Sarcoma Herpes Virus (KSHV) or Epstein–Barr virus, it has been shown that Gly can interfere with the activity of their viral oncoproteins. On one hand, Gly suppresses the expression of the viral oncoproteins E6 and E7 from HPV [8] and LANA from KSHV [9]. In the cases of HPV and KSHV, suppression of their viral oncoproteins leads to p53 reactivation, increased reactive oxygen species (ROS) and mitochondrial dysfunction, resulting in G1 cell cycle arrest, DNA fragmentation and oxidative stress-mediated apoptosis [8,9]. As for the suppression of the EBV viral oncoprotein LMP1, Gly inhibits the SUMOylation processes derived from LMP1 activity, blocking proliferation, increasing cell death, inducing low levels of viral reactivation and preventing the infection of new cells [10].

Other mechanisms of antiviral action have been described for Gly against other viruses, and it has not been ruled out that some of these may also be involved in its action against HPV infectious processes. It has been observed that Gly can be effective since the steps prior to infection, directly inactivating the viruses or preventing their binding and entry into the host cell, until the infection is established by inhibiting viral replication, stimulating apoptosis, or even modulating the immune response [4,6,7]. Specifically, it has been observed that Gly can directly inactivate HSV-1 and VZV viruses [11,12,13]. Gly is also able to prevent the binding of SARS-CoV-2 and hepatitis and influenza viruses to the cell surface by interfering with ACE2 and HMGB1 [14,15,16] or to reduce cell membrane and viral envelope fluidity by decreasing the HIV intercellular fusion [17]. Once the viruses have already penetrated the host cell, Gly is also able to inhibit the replication of HSV-1, influenza virus, HIV, SARS-CoV, SARS-CoV-2, rotavirus or Zyka [14,15,18,19,20,21,22,23]. And once the virions have assembled, it has also been shown, in the case of HCV, that Gly can reduce its release [24] (Figure 2).

## 3. Anticancer Activity of Glycyrrhizinic Acid

Gly has been shown to have onco-preventive and onco-therapeutic properties against different types of cancer. In fact, Gly has been shown in preclinical models to inhibit the onset and development of different types of cancer such as cervical, endometrial, colon, gastric, leukaemia, glioblastoma, lung, liver and melanoma. At the same time, it also has a synergistic action with different chemotherapeutics and is useful in reducing the adverse effects of chemotherapy, including hepatotoxicity, nephrotoxicity, genotoxicity, neurotoxicity and pulmonary toxicity [3,5,8,25,26].

The different mechanisms of anticancer action described for Gly include antiproliferative, proapoptotic, anti-invasive/antimetastatic, immunomodulatory, anti-inflammatory and antioxidant action. Gly may also inhibit the development of oestrogen-dependent cancers such as endometrial cancer, although it is unclear whether this may be due to competitive inhibition of the oestrogen receptor or to the repression of the expression of pro-inflammatory enzymes and cytokines. These anti-cancer properties are thought to be mediated via the regulation of different cell signalling pathways by Gly. Among the pathways modulated by Gly are NF-κB, MAPK, PI3K/Akt/mTOR, Akt/mTOR/STAT3, PKC/ERK, and Notch [3,8,25]. On the other hand, Gly is also able to inhibit oncogenesis mediated by viral oncoproteins such as E6 and E7 of HPV, LANA of KSHV, or LMP1 of EBV [8,9,10] (Figure 3).

## 4. Anticancer Action of Glycyrrhizinic Acid in Cervical Cancer Models

The antiproliferative and proapoptotic action of Gly has been evaluated in different cervical cancer cell models such as HeLa, Ca Ski and C-33 A cells. HeLa cells are cervical epithelial cells obtained from a 31-year-old black female cervical adenocarcinoma containing HPV-18 sequences and with low p53 expression and normal pRB levels [27]. Ca Ski is an advanced cervical cancer cell line isolated from cells from a small bowel mesentery metastasis of a 40-year-old white woman with squamous cell carcinoma and containing integrated HPV-16 genome (about 600 copies per cell) as well as HPV-18-related sequences [28]. C-33 A is a cervical cancer cell line extracted from a 66-year-old white woman characterised by elevated p53 expression, normal pRB levels and negative for HPV DNA and RNA [29]. The use of C-33-A cells is of interest because it provides insight into the properties of Gly on a comprehensive phenotype of cervical cancer cells in the absence of HPV.

### 4.1. Inhibition of the Expression of the Viral Oncoproteins HPV E6 and HPV E7

The E6 and E7 oncoproteins encoded by HPV are the main agents causing the cervical cancer. In addition, these viral oncoproteins have previously been reported to have an effect on p53 and pRb by increasing their ubiquitin-mediated degradation. The effect of Gly treatment on the expression of viral oncoproteins E6 and E7 has been evaluated in Ca Ski cervical cancer cells. Under the conditions tested, Gly reduced the expression of HPV E6 by up to 71% (0.29 ± 0.08 folds) and HPV E7 by up to 82% (0.18 ± 0.04 folds). In addition, it was also observed that this inhibition of viral oncoprotein expression was associated with a restoration of p53 levels, the expression of which was increased by up to 516% (5.16 ± 0.47 folds) as compared to the control under the conditions tested [8].

### 4.2. Antiproliferative Action of Glycyrrhizinic Acid

Farooqui et al. studied for the first time the antiproliferative and proapoptotic action of Gly in a cervical cancer cell model [26]. In their study, they used the HeLa cell line of HPV18^+^ cervical cancer and evaluated the effect on viability and proliferation of cervical cancer cells via treatment with increasing doses of Gly at different times. The results showed that Gly inhibited the viability and proliferation of HeLa cells with a cytotoxic effect on HeLa cells in a concentration- and time-dependent manner. In parallel, Gly-treated HeLa cells were evaluated via flow cytometry to see if its inhibitory effect on cell growth in these cervical cancer cells was due to the interruption of cell cycle progression. The results showed that under normal conditions, HeLa cells had the following cell cycle distribution: 49.19% in G0/G1 phase, 22.34% in S phase and 28.47% in G2/M phase. The Gly treatment, meanwhile, resulted in an accumulation of G0/G1 phase cells and reduction of S and G2/M phase cells. In addition, the appearance of a sub-G0/G1 population (hypodiploid cells) was observed in the histogram indicating the accumulation of apoptotic and dead cells. This effect was dose-dependent, reaching 5.26%, 8.40%, 12.62%, 14.79% and 37.37% of apoptotic cells after treatment with 20, 40, 80, 160 and 320 μM of Gly, respectively. On the other hand, to rule out that Gly also had a cytotoxic action on healthy cells, they evaluated the effect of the same doses of Gly on a normal colon epithelial cell line (IEC-6), and found that Gly had a negligible effect on these cells. Moreover, they also observed that Gly has a synergistic action with cisplatin and 5-FU in blocking cell proliferation.

Based on these results, Farooqui et al. conclude that treatment with Gly alone reduces the viability of HeLa cells and has a synergistic effect with cisplatin and 5-FU on the viability of this cervical cancer cell line, potentially reducing the necessary doses of these drugs to achieve the same effect. Furthermore, they point out that Gly has cytotoxic effect on cervical cancer cells without a cytotoxic effect on healthy cells at the same dose [26].

Moreover, Ahmad et al. have studied the antiproliferative and proapoptotic properties of Gly in cell lines of HPV-16^+^ cervical cancer (Ca Ski cells) and even in a non-HPV cervical cancer line (C-33 A) [8,25]. Similarly to the cell line of HPV-18^+^ cervical cancer, Gly is able to inhibit cell growth and proliferation by stopping the cell cycle in the G0/G1 phase in both cell lines. Furthermore, the effect was also dose- and time-dependent and no cytotoxicity on normal cells was observed.

### 4.3. Proapoptotic Action of Glycyrrhizinic Acid

To establish whether the antiproliferative property of Gly observed in cervical cancer cells (HeLa, Ca Ski and C-33 A) was due to apoptosis, respective fluorescence-activated cell sorting (FACS)-based Annexin-V FITC/PI trials were performed to measure the amount of apoptosis in different cervical cancer cells treated with different doses of Gly. The results of these trials showed that Gly treatment significantly reduced the number of viable cells and increased the number of apoptotic cells in a dose-dependent manner [8,25,26].

Apoptosis is characterised by prominent nuclear changes in a cell, such as nuclear fragmentation and condensation. In this regard, Gly induces nuclear condensation and fragmentation in a dose-dependent manner in both HeLa and Ca Ski cells as well as in C-33 A cells, which is indicative of its apoptosis-inducing effect in different cervical cancer cells [8,25,26].

The proapoptotic action of Gly in cervical cancer cells is thought to be mediated by both the intrinsic and extrinsic pathways of apoptosis [8,25,26]. Caspases-8 and -9 are the initiator caspases of the extrinsic and intrinsic apoptotic pathways, respectively, and caspase-3 is the main effector caspase of both pathways. The treatment of HeLa and C-33 A cervical cancer cell lines with Gly results in activation of caspases 3, 8 and 9, which means it induces apoptosis via both the intrinsic and extrinsic pathways. However, when these cells were pre-treated with caspase inhibitors, this did not result in complete blockage of Gly apoptotic activity, indicating that the apoptotic action of Gly may also involve activation of other caspase-independent apoptotic pathways [25,26]. Similarly, Gly also activates caspases 3, 8 and 9 in Ca Ski cervical cancer cells. However, pretreatment with inhibitors of these caspases did completely annul Gly-mediated apoptosis in Ca Ski cells, so, in this case, the apoptotic action of Gly would be mainly mediated by caspases [8].

The treatment of the different models of cervical cancer with Gly results in a reduction in the mitochondrial membrane potential (ΔΨm), which is the characteristic feature of the activation of the intrinsic or mitochondrial apoptotic pathway [8,25,26]. In addition, studies in Ca Ski and C-33 A cell lines also assessed the mRNA levels of Bax, Bad (pro-apoptotic genes) and Bcl-2 (anti-apoptotic genes) that critically regulate the mitochondria-mediated intrinsic pathway. In both studies, Gly treatment was found to increase the expression of the pro-apoptotic genes Bax and Bad and reduce the expression of the anti-apoptotic gene Bcl-2 [8,25].

### 4.4. Intracellular Increase in Reactive Oxygen Species (ROS)

The generation of ROS is considered a primary signal of the apoptosis onset. Both caspase-dependent and caspase-independent forms of cell death are activated by increased production of intracellular reactive oxygen species (ROS) [26,30]. It has been observed that the treatment of HeLa cells with Gly increases ROS formation, and that pretreatment with ROS formation inhibitors blocked ROS formation and reduced Gly cytotoxicity, although not completely [26]. Equivalent results were obtained in C-33 A and Ca Ski cells, where a dose-dependent increase in intracellular ROS levels was also observed after Gly treatment. Similarly to what was observed in HeLa cells, the pretreatment of C-33 A and Ca Ski cells with an ROS inhibitor completely blocked ROS formation, and although in these cases the reduction in cytotoxicity was even greater, it also did not completely reduce ROS formation [8,25]. This indicates that increased intracellular ROS is a crucial step for Gly-mediated apoptosis in cervical cancer cells, but that additional apoptotic pathways may also be involved [8,25,26].

Glutathione (GSH) is an antioxidant tripeptide that is important in protecting against cell damage induced by excess ROS in both healthy and cancer cells. In addition, elevated glutathione levels correlate with increased proliferation within various cell types and are generally at higher levels in cancer cells compared to non-cancer cells. So, to make a better profile of the effect of Gly treatment on oxidative stress, glutathione levels were also quantified in Ca Ski cell experiments, and it was observed that Gly treatment substantially reduced total glutathione levels. Therefore, these findings suggest that Gly-mediated induction of apoptosis in CaSki cervical cancer cells is in line with the notion that GSH depletion promotes programmed cell death [8].

### 4.5. Inhibition of the Notch Pathway

The Notch signalling pathway is initiated by the binding of Notch ligands, such as Delta and Jagged, to the Notch receptor. Binding initiates the split of Notch receptor, activating the Notch intracellular domain (NICD). This activation, in turn, can activate pathways such as PI3K-Akt. In addition, intracellular Notch moves into the nucleus. Here, it initiates the transcriptional activation of Notch1 target genes, including HES1, cyclin D1, c-MYC, etc. [31].

Abrupt activation of the Notch pathway has been described in several human malignancies, including breast, lung and cervical cancer, among others. In addition, elevated Notch-1 receptor levels have been directly correlated with the progression of cervical intraepithelial neoplasia (CIN) to metastatic cervical carcinoma [32,33]. Ahmad et al. observed in studies that GA treatment inhibits the Notch pathway through the inhibition of Notch-1 and Jagged-1 in different cervical cancer cell lines (Ca Ski and C-33 A). In addition, Gly inhibited the expression of Hes-1 mRNA, which is a crucial downstream target of Notch-1 [8,25]. On the other hand, cyclins and their dependent kinases, also known as CDKs (Cyclin-Dependent Kinases), are responsible for the homeostatic regulation of the cell cycle, with inactivation of CDKs being responsible for cell cycle stoppage. Cyclin D1 is also another important signalling protein of the Notch pathway that is activated by the NICD. It has been observed that increased levels of cyclin D1 expression in HPV-mediated cervical adenocarcinoma cells can lead to a reduction in time within the G1 phase, which subsequently results in cell proliferation [34]. Studies in C-33 A and Ca Ski cervical cancer cells showed that Gly treatment resulted in increased expression of p21, an inhibitor of CDK activity. Accordingly, it was observed that Gly treatment of C-33 A and Ca Ski cells inhibited cyclin D1 and CDK4 expression, which was associated with growth suppression and G0/G1 cell cycle stoppage [8,25].

In addition, a correlation was established between Notch-1 signalling and Bcl-2 family members in cervical cancer cells. Specifically, inhibition of the Notch-1 signalling pathway by Gly treatment was found to be responsible for activation of pro-apoptotic protein expression (Bad and Bax) and repression of anti-apoptotic protein expression (Bcl-2) [25].

These data suggest that induction of apoptosis and growth suppression in Gly-treated cervical cancer cells is associated with inhibition of the Notch pathway.

### 4.6. Regulation of the Tumour Microenvironment

HPV infection is not sufficient to develop cancer; other host and environmental factors must also be involved. Among them, it has been proposed that cervical metaplasia may create an immunosuppressive environment that favours malignant transformation [35]. In fact, it is known that transformed cells can secrete immunosuppressive molecules including the HMGB1 (high-mobility group box-1) cytokine. The oncogenic action of HMGB1 is HPV-independent. However, HMGB1 secretion during cervical cancer progression impairs the antiviral response by inhibiting IFNα secretion and increasing ICOSL (Inducible T Cell Costimulator Ligand) expression by plasmacytoid dendritic cells (pDCs). This may promote the accumulation of Treg cells in the tumour environment and lead to the development of a tolerogenic microenvironment [36]. Gly has been shown to be able to block HMGB1 by direct binding to it, both in cervical cancer cells and in other tissues and in animals. In addition, intraperitoneal injection of Gly into immunocompetent mice has been shown to produce extracellular blockage of HMGB1, activating the anti-cancer immune response and inhibiting the growth of pre-established solid tumours [37].

Anticancer properties of glycyrrhizinic acid in cervical cancer models is reviewed in Figure 4.

## 5. Clinical Studies of Glycyrrhizinic Acid in HPV Patients

Gly administered both topically and orally has been evaluated in different clinical studies in patients with HPV and/or LSIL in the cervix, vagina or vulva, as well as in women and men with anogenital condylomas and in patients with focal multi-epithelial hyperplasia in the oral cavity. A total of nine studies have been reported, involving a total of 531 patients. In terms of methodology, one study was randomised and placebo-controlled, three studies compared against another treatment, and there were five prospective uncontrolled observational studies (Table 1). All the published studies have been conducted with the same topical product or combination of topical and oral products under the brand name Glizigen^®^ (Catalysis S.L., Toledo, Spain). Glizigen^®^ spray, also referred to in some publications as Epigen^®^ spray, contains 0.1% glycyrrhizinic acid. Oral Glizigen^®^, referred to in publications as Viusid^®^, is a glycyrrhizinic acid, L-arginine, L-glycine, vitamin C, B5, B6, B9, B12 and zinc-based nutritional supplement.

The first study in women with cervical LSIL evaluated the topical use of Gly for 10 days, with lesions normalising in 80% of cases by day 30 [38]. Subsequent studies evaluated generally longer treatment times up to 8–12 weeks. The second study compared the efficacy of local action of Gly versus imiquimod in women with LSIL on cervical-vaginal cytology, with Gly showing superior efficacy (complete histological remission in 57% of cases treated with Gly vs. 18% in those treated with imiquimod). In addition, Gly showed fewer adverse effects both locally (7% Gly vs. 62% imiquimod) and systemically (0% Gly vs. 38% imiquimod) [39]. The following two studies evaluated the combination of topical and oral Gly treatment. The first study found that treatment of LSIL lesions for 12 weeks led to negative cytology in 74% of cases [40]. In the other study, combined topical and oral treatment with Gly was given to both the HPV-positive woman and her sexual partner, and in this case, HPV negativisation in the patients was observed even faster and with greater magnitude (88.8% at 4 weeks and 100% at 8 weeks). This study also assessed cases of recurrences after 10–15 months, which occurred in 14.8% (4/27) of cases. Interestingly, these four recurrences occurred within a group of 5 (4/5, 80%) patients who reported that their sexual partners either did not undergo treatment or had a second sexual partner who had not been treated [41].

The efficacy of topical and oral Gly against anogenital condyloma in women and men has also been demonstrated in two clinical studies compared to placebo and podophyllotoxin. The double-blind study compared to placebo in boys and girls with condylomas provides very clear results after 8 weeks of treatment, showing zero efficacy with placebo (0%) compared to 68.4% of complete elimination of condylomas observed with Gly [42]. On the other hand, combined topical and oral therapy with Gly has also shown superior efficacy to podophyllotoxin, with complete elimination of condyloma of 87.5% in patients treated with Gly vs. 76% in those treated with podophyllotoxin. At the same time, the safety of Gly is higher, with mild adverse effects such as itching observed in 18% of patients treated with Gly, while patients treated with podophyllotoxin suffered more frequent and more severe adverse effects in 46% of cases [43].

The efficacy and safety of Gly during pregnancy has also been evaluated in patients with HPV. On one hand, Gly has been used for the treatment of anogenital condylomas in pregnant women with no observed adverse effects and with an efficacy of 80% for the complete elimination of small condylomas (≤5 mm) and 53.3% for large condylomas (>5 mm) [44]. Intravaginal application of Gly during the last week of pregnancy has also been evaluated in HPV-positive women, with HPV negative results and elimination of vaginal papillomatosis after delivery. In addition, none of the babies born tested positive for HPV at day 1 of life or after 1.5 years of follow-up [45].

Finally, Gly has also been evaluated in patients with focal multi-epithelial hyperplasia in the oral cavity due to HPV. This study showed 63% efficacy after 4 weeks of treatment with topical Gly, and 81% efficacy after 12 weeks of treatment with liquid nitrogen. It is worth noting that the efficacy observed with Gly, although lower than with liquid nitrogen, was obtained with a shorter treatment time (4 vs. 12 weeks) and had fewer adverse effects [45].

## 6. Discussion

Human papillomavirus is an oncogenic virus, so addressing HPV infections by using agents with both antiviral and anticancer properties can be a comprehensive therapeutic and preventive option. This paper has reviewed the available preclinical and clinical evidence on the antiviral and antiproliferative properties of glycyrrhizinic acid to evaluate its potential use in the treatment of HPV infections and in the prevention of the development of HPV-associated lesions via topical and/or oral administration.

The antiviral properties of Gly have been described in multiple studies, having shown broad-spectrum antiviral action against both human and animal viruses. Human viruses for which Gly has shown antiviral action include oncogenic viruses such as human papillomavirus (HPV), Kaposi’s sarcoma herpesvirus (KSHV) and Epstein–Barr virus (EBV), as well as other viruses such as human immunodeficiency virus (HIV), herpes simplex virus (HSV), coronavirus, influenza virus, etc. [4,6,7].

The mechanisms of antiviral action described against these viruses cover most phases of the virus replication cycle, having shown antiviral activity from pre-infection steps, by directly inactivating the viruses or preventing their binding and entry into the host cell, until after the infection, by inhibiting viral replication, stimulating apoptosis or even modulating the immune response [4,6,7,11,12,13,14,15,16,17,18,19,20,21,22,23,24]. This does not mean that for a given virus Gly will act at all these levels, but it cannot be ruled out that for a virus for which a specific action has been described, other mechanisms may also be involved that have not been described so far.

In the specific case of antiviral action against HPV, the ability of Gly to inhibit the expression of the viral oncoproteins E6 and E7, which are involved in the process of cell proliferation and transformation, has been described [8]. This antiviral action of Gly against HPV would also be part of its anticarcinogenic mechanism of action, since the viral oncoproteins E6 and E7 are the main agent causing cervical cancer by deactivating the tumour suppressors p53 and pRb.

The anticancer action of Gly on cervical cancer cells is also thought to be mediated via the inhibition of the Notch signalling pathway [8,25], which is linked to the progression of cervical intraepithelial neoplasia (CIN) to metastatic cervical carcinoma [32,33]. In addition, Gly induces an increase in reactive oxygen species (ROS) in cervical cancer cells. As a result of these properties, Gly exerts an antiproliferative action and induces apoptosis of cervical cancer cells by both intrinsic and extrinsic activation of apoptosis [8,25,26].

Moreover, Gly also contributes to regulating the tumour microenvironment by directly binding and blocking HMGB1 (high-mobility group box-1), an immunosuppressive cytokine that helps transformed cells evade the immune system [37].

Finally, the different clinical studies conducted with Gly administered both topically and orally provide evidence of its safety and efficacy in men and women, including during pregnancy, in different HPV-derived processes such as infections and lesions in the cervix, vagina and vulva, anogenital condylomas or focal multi-epithelial hyperplasia in the oral cavity [38,39,40,41,42,43,44,45,46].

Among the different clinical studies analysed in this review, superior efficacy seems to be observed when combined therapy is used locally and systemically, although we cannot make that statement definitively. On the other hand, the treatment times carried out in the different studies vary from 10 days to 12 weeks. Everything indicates that the 8-week treatment period may be the time necessary to achieve optimal efficacy without significant differences with longer treatment periods that may reduce treatment adherence. In order to strengthen this evidence, this research team has recently launched an RCT that evaluates the effectiveness of combined local and systemic treatment with glycyrrhizinic acid over a period of 8 weeks (NCT05916911).

With regard to limitations of this review, the mechanisms of antiviral and anticancer action of Gly may not yet be fully defined. Although multiple mechanisms of action have been described, some remain to be elucidated, and future research may help to understand these mechanisms of action even more fully. On the other hand, most of the clinical studies conducted with Gly are observational and involve a small population. However, the availability of one randomised study compared to placebo and other studies comparing Gly to another treatment provide a higher quality of evidence. This supports the need for further randomised controlled studies of samples that are sufficient to support Gly’s therapeutic potential for HPV-associated problems.

## 7. Conclusions

The scientific evidence reviewed on glycyrrhizinic acid highlights its broad-spectrum action as an antiviral and anticancer agent and defines the mechanisms and specifics observed against HPV and HPV-derived oncogenic processes. At the same time, glycyrrhizinic acid is proposed as a potential therapeutic agent against HPV infections and lesions.

## Figures and Tables

**Figure 1 jpm-13-01639-f001:**
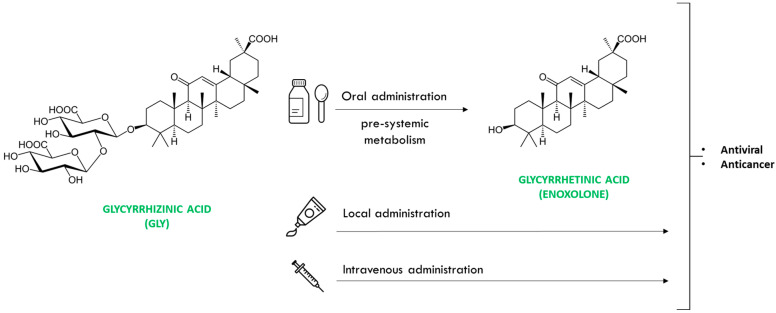
Chemical structure of glycyrrhizinic acid, routes of administration and drug action.

**Figure 2 jpm-13-01639-f002:**
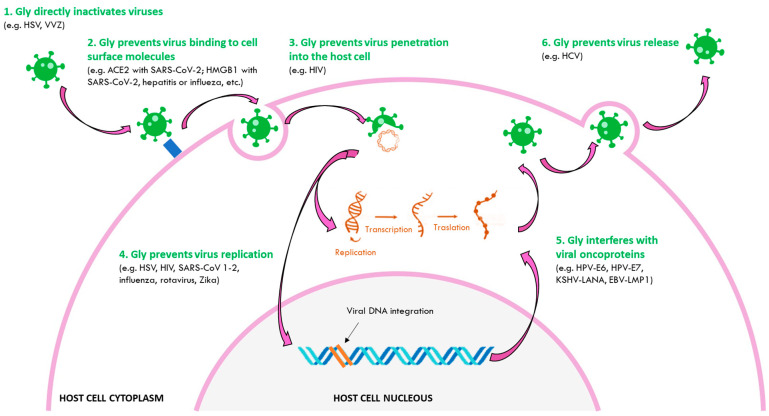
Antiviral mechanisms of glycyrrhizinic acid. Gly directly inactivates some viruses (e.g., HSV, VZV), prevents the binding of viruses to cell surface molecules (e.g., ACE2 in SARS-CoV-2 or HMGB1 in SARS-CoV-2, hepatitis or influenza, etc.), prevents the penetration of the virus into the host cell (e.g., HIV), prevents virus replication (e.g., HSV, SARS-CoV 1 and 2, influenza, rotavirus, Zika), interferes with the expression or activity of viral oncoproteins (e.g., HPV-E6, HPV-E7, KSHV-LANA, EBV-LMP1), and prevents the release of viruses (e.g., HCV).

**Figure 3 jpm-13-01639-f003:**
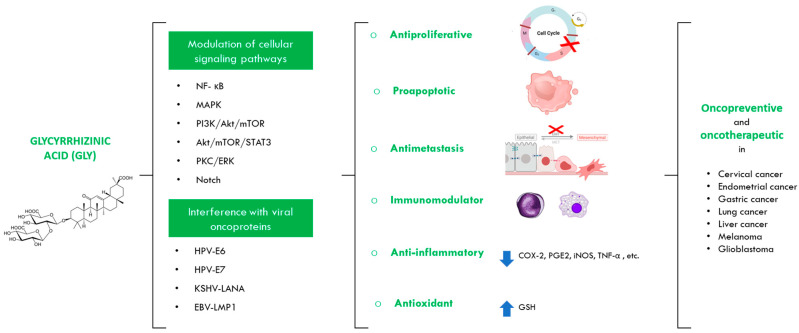
Anticancer properties of glycyrrhizinic acid. Gly has oncopreventive and oncotherapeutic properties against different cancer models by modulating different cellular signaling pathways and interfering with viral oncoproteins that lead to an antiproliferative, proapototic, antimetastatic, immunomodulatory, anti-inflammatory and antioxidant action. (Red X’s mean that Gly blocks those steps; down and up arrows respectively mean that Gly represses or induces those proteins).

**Figure 4 jpm-13-01639-f004:**
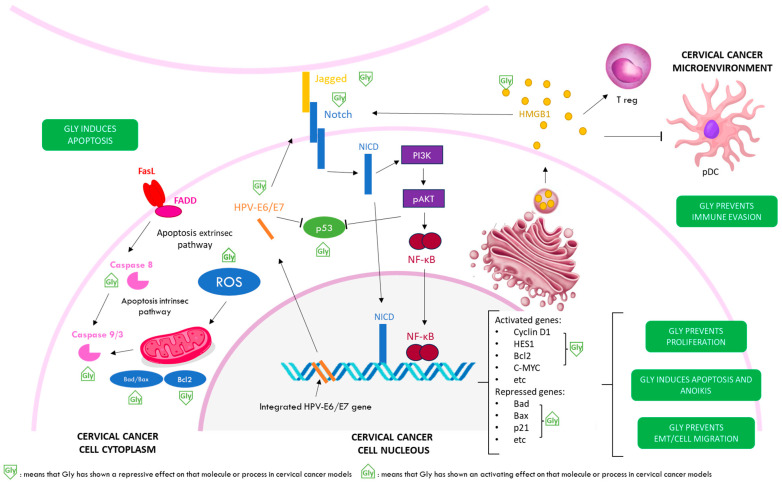
Anticancer properties of glycyrrhizinic acid in cervical cancer models. Gly inhibits the Notch signalling pathway, leading to reduced expression of Clicin D1, HES1, Bcl2, C-MYC, etc., and to an increase in the expression of Bad, Bax, p21, etc. As a result, Gly prevents the proliferation of cervical cancer cells, induces their apoptosis, and prevents epithelial–mesenchymal transition and cell migration. The proapototic action of Gly is mediated by both the intrinsic and extrinsic pathways. Gly also reduces the levels of the viral oncoproteins HPV-E6/E7, leading to an increase in p53. On the other hand, Gly directly inactivates HMGB1, thus preventing the increase in Treg cells and a decrease in pDCs in the tumour microenvironment, preventing evasion to the control of the immune system.

**Table 1 jpm-13-01639-t001:** Summary of clinical studies with glycyrrhizinic acid in patients with HPV.

Ref.	Diagnosis	Type of Study	Participants	Intervention	Results
[38]	Cervical LSIL	Observational	97 women	Glizigen^®^ topical 10 days	Normal cytology in 80% of cases 20 days after completion of treatment.
[39]	Cervical-vaginal LSIL	Comparative	83 women	Glizigen^®^ topical 2–6 weeks vs. imiquimod 8 weeks	Local adverse effects: imiquimod 62% vs. 7% Gly. Systemic adverse effects: imiquimod 38% vs. 0% Gly. Full histological remission: imiquimod 18% vs. 57% Gly
[40]	HPV infection/LSIL in cervix	Observational	62 women	Glizigen^®^ for topical and oral use 8 weeks	74% HPV negative at 12 weeks and 100% HPV negative at more than 13 weeks. LSIL elimination in 73% at 12 weeks.
[41]	Cervical HPV infection	Observational	27 women and their respective sexual partners	Glizigen^®^ topical and oral 4–8 weeks	88.8% of negativization of HPV in one month. 100% of negativization of HPV in two months. 14.8% (*n* = 4) relapses after 10–15 months.
[42]	Anogenital condylomas	Double-blind compared to placebo	76 children	Glizigen^®^ topical and oral vs. placebo 8–12 weeks	Complete elimination of 68.4% with Gly vs. 0% with placebo.
[43]	Anogenital condylomas	Comparative	100 men and women	Glizigen^®^ for topical and oral use 8 weeks vs. podophyllotoxin 6 weeks	Complete elimination of 87.5% with Gly vs. 76% with podophyllotoxin. 18% of adverse effects with Gly vs. 46% with podophyllotoxin.
[44]	Anogenital condyloma during pregnancy	Observational	40 women	Glizigen^®^ topical 3 weeks (combined with electrocoagulation in condyloma ≤ 5 mm)	Complete elimination of 80% of small condyloma (≤5 mm) and 53.3% of large condyloma (>5 mm).
[45]	Cervical HPV infection/vaginal papillomatosis during pregnancy	Observational	26 women	Glizigen^®^ topical 1 week before delivery	100% of newborns had negative PCR for HPV at day 1 of life and after 1.5 years. 100% (11/11) elimination of vaginal papillomatosis after childbirth with no recurrences after 1.5 years. 100% (15/15) negativization of HPV after delivery, with 40% recurrences at 4 months.
[46]	Focal multi-epithelial hyperplasia in the oral cavity	Comparative	20 men and women	Glizigen^®^ topical 4 weeks vs. liquid nitrogen 12 weeks	Gly showed 63% efficacy after 4 weeks, while liquid nitrogen showed 81% efficacy after 12 weeks. Gly had fewer adverse effects.

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
