# Peer review of "Glycyrrhizinic Acid as an Antiviral and Anticancer Agent in the Treatment of Human Papillomavirus"

_jpm, 2023, doi:10.3390/jpm13121639_

Round 1

Reviewer 1 Report

Comments and Suggestions for Authors

This manuscript summarizes the molecular properties of Glycyrrhizinic acid, a pharmaceutical agent targeting HPV-related lesions, widely available in several parts of the world. With a clear statement regarding funding and COI’s, this submission's conclusions should be regarded with caution.

The manuscript indeed outlines thoroughly most known and proposed mechanisms of Glycyrrhizinic acid action against low grade cervical lesions and highlights the several known unknowns. In this regard it represents a comprehensive review.

My main suggestions are:

1. Since this agent is usually prescribed by clinicians, the subsection “5. Clinical studies of glycyrrhizinic acid in HPV patients” needs more development. Large randomized RCT’s on Glycyrrhizinic acid are inexistent, which is disappointing for an expensive medication which is available in the market for several years. Of course, this also applies for most other agents claiming local antiviral properties against low grade cervical dysplasias; mainly because of well documented intrinsic difficulties to design epidemiologically state of the art studies for a self-restricted condition (lower anogenital tract LSIL’s).  

2. Furthermore, for a compound with hefty out of the pocket costs, it is paramount to standardize a dosage protocol, either for local, systematic or combined administration. This should be made clearer in the manuscript’s text. Failing to standardize a dosage scheme leads to increased patient drop-outs, especially for a costly medication and subsequently to an inability to assess efficacy and effectiveness.

3. Of particular interest are the paragraphs addressing safety in pregnancy issues, as well as a potential use of this agent against the upcoming pandemic of oral HPV infection.

Comments on the Quality of English Language

Minor language polishing might be required.

Author Response

Dear reviewer,

I greatly appreciate the time you have dedicated to reviewing this manuscript and your comments, for which I agree with you on all of them. Below you can find the answer to your suggestions one by one:

  1. Some medium-sized RCT has been perfomed comparing glycyrrhizinic acid with placebo or other therapeutic agents (e.g. podophyllotoxin or imiquimod). However, we are aware of the need to carry out more and larger-scale studies. For this reason, this research team has launched an RCT and we are aware that other studies are being developed. To clarify this, we have included in the manuscript a mention of the study we are carrying out along with the clinicaltrials.gov identifier number (NCT05916911).
  2. Among the different clinical studies analyzed in this review, superior efficacy seems to be observed when combined therapy is used locally and systemically, although we cannot make that statement definitively. On the other hand, the treatment times carried out in the different studies vary from 10 days to 12 weeks. Everything indicates that the 8-week treatment period may be the time necessary to achieve optimal efficacy without significant differences with longer treatment periods that may reduce treatment adherence. For this reason, the clinical study launched by this research team aims to evaluate the effectiveness of combined local and systemic treatment with glycyrrhizinic acid over a period of 8 weeks. To clarify this, we have included in the manuscript a mention of the duration of the local and systemic treatment regimen that will be evaluated in our clinical study.
  3. Thanks for this appreciation. To the best of our knowledge, it is the first evidence of the use of any natural therapeutic agent used in these cases with unmet therapeutic needs, so new avenues of research in this direction will be of great interest.

In relation to the study carried out in patients with oral HPV infection, we have recently been able to confirm with the researcher who carried out this study the formula of glycyrrhizinic acid that was used, so we included this specification in the manuscript.

Attached you will find the revised manuscript with the changes made highlighted. I hope that my comments and the changes made to the manuscript fully respond to your suggestions.

Sincerely,

Reviewer 2 Report

Comments and Suggestions for Authors

This interesting  work  gives a  scientific overview  about glycyrrhizinic acid highlights, including  its broad-spectrum action as an antiviral and anticancer agent and defines the mechanisms and specifics effects  against HPV and HPV-derived oncogenic processes.  According with results obtained in observational clinical triales in patients with HPV infection, potential use in the treatment of precancerous and cancerous lesions is highlighted. However, the authors  discuss some limitations related with the mechanisms of antiviral and anticancer action of  Gly  which are not completely dilucidated, and the  and the lack of randomised studies  compared to placebo and others comparing Gly to another treatment  that can provide a higher quality of evidence.

Author Response

Dear reviewer,

I greatly appreciate the time you have dedicated to reviewing this manuscript and your comments, for which I agree with you on all of them. I want to thank you for your appreciations about the broad-spectrum action of glycyrrhizinic acid as an antiviral and anticancer agent. Regarding clinical trials, some medium-sized RCT has been perfomed comparing glycyrrhizinic acid with placebo or other therapeutic agents (e.g. podophyllotoxin or imiquimod). However, we are aware of the need to carry out more and larger-scale studies. For this reason, this research team has launched an RCT and we are aware that other studies are being developed by other research teams. To clarify this, we have included in the manuscript a mention of the study we are carrying out along with the clinicaltrials.gov identifier number (NCT05916911).

Attached you will find the revised manuscript with the changes made highlighted in response to al reviewers. I hope that my comments and the changes made to the manuscript fully respond to your suggestions.

Sincerely,